# Phytochemical and Pharmacological Evaluation of the Residue By-Product Developed from the *Ocimum americanum* (*Lamiaceae*) Postdistillation Waste

**DOI:** 10.3390/foods10123063

**Published:** 2021-12-09

**Authors:** Izabela Jasicka-Misiak, Mariia Shanaida, Nataliia Hudz, Piotr Paweł Wieczorek

**Affiliations:** 1Department of Pharmacy and Ecological Chemistry, University of Opole, 45-052 Opole, Poland; izabela.jasicka@uni.opole.pl (I.J.-M.); natali_gudz@ukr.net (N.H.); 2Department of Pharmacognosy and Medical Botany, I. Horbachevsky Ternopil National Medical University, 46-001 Ternopil, Ukraine; 3Department of Drug Technology and Biopharmacy, Danylo Halytsky Lviv National Medical University, 79-010 Lviv, Ukraine; 4Department of Analytical Chemistry, University of Opole, 45-052 Opole, Poland; pwiecz@uni.opole.pl

**Keywords:** American basil, herb, hydrodistilled residue by-product, safety, polyphenols, antioxidant activity, anti-inflammatory activity

## Abstract

The yield of essential oils in plants is not high and postdistillation wastes rich in phenolic compounds could be used to enhance the profitability of essential oil-bearing plants. The aim of the study was to evaluate polyphenols in a dry extract obtained from the postdistillation waste of the American basil (*Ocimum americanum* L.) herb, and to conduct the screening of its pharmacological activities. Rosmarinic acid, caffeic acid and rutin were identified in the extract using high-performance thin-layer chromatography. The high-performance liquid chromatography analysis found the presence of a plethora of polyphenols in the extract. Rosmarinic acid, luteolin-7-*O*-glucoside and rutin were as the main compounds. The total phenolic content in the extract was 106.31 mg GAE/g and free radical scavenging activity against 2,2-diphenyl-1-picrylhydrazyl evaluated as IC_50_ was 0.298 mg/mL. The tested extract dose-dependently decreased the paw edema in rats, suggesting its potent anti-inflammatory property. The acute toxicity study indicates its safety. Thus, the *O. americanum* hydrodistilled residue by-product is the promising source of biologically active compounds with significant antioxidant and anti-inflammatory effects.

## 1. Introduction

Plant raw materials and herbal preparations are traditionally administered for the treatment or prevention of many diseases from ancient times [1]. Natural compounds, which are synthesized by plants, are structurally ‘optimized’ in the process of evolution to serve the regulation of their defense and competition with other living organisms. It explains their ability to fight ailments of the human body, such as infectious diseases, cancer, chronic inflammation, etc. [2].

The quality of herbal preparations is mainly guaranteed by the defined process of their preparation, which involves the use of plant raw materials of appropriate quality [3,4]. In recent decades, the distribution areas of many wild species of medicinal plants get less due to significant anthropogenic load. It leads to the depletion of the sources of the raw materials and reflects the necessity of finding new promising species of plants that are able to accumulate the targeted bioactive compounds.

Aromatic medicinal plants represent a valuable source of herbal drugs. The genus Basil (*Ocimum* L.) belonging to the *Nepetoideae* Burnett subfamily of the *Lamiaceae* family includes 76 species of essential oils-bearing plants [5,6]. Most of them are native to Eurasia and Africa. Several species (*O. basilicum*, *O. sanctum*, *O. tenuiflorum*) are successfully cultivated on almost all continents.

American basil (*O. americanum* L.) originated from Africa where it is locally used for treating disorders of the digestive system, respiratory tract and as a sedative drug [7,8]. The water extracts of *O. americanum* aerial parts were orally administered for treating cough, bronchitis, immunity disturbance, dysentery, and as a mouth wash for reliving toothache [8]. This species was naturalized in America and nowadays it is gradually spreading in Europe. Despite the presence of some experimental studies regarding the chemical composition and biological activity of *O. americanum* [9,10,11,12], this species is of considerable scientific interest. Essential oil hydrodistilled from its herb contained linalool as a predominant compound possessing marked sedative effect [11]. Six different chemotypes of *O. americanum* were found in the natural African and Asian habitats of this species [8]. Vieira et al. [8] supposed that these chemotypes may have originated through introgressive hybridization with other *Ocimum* species. The presence of a significant number of chemotypes within *O. americanum* due to the large area of its wild distribution [8] suggests that its cultivation for obtaining raw material under controlled conditions is quite a perspective.

A lot of waste rich in phenolic compounds can be obtained due to the rational processing of the raw materials of aromatic plants [4,13]. The recovery of fractions with prospective biological potential from residues that otherwise they would be lost is a very economically important issue. As it is known, the amount of essential oils distilled from herbs or leaves is not higher than 3% [13,14]. Hydrodistillation residues in significant quantities remain after the isolation of volatile compounds from essential oil-bearing plants [4,13,15,16,17,18]. It allows obtaining polyphenols and other non-volatile pharmacologically active substances from the residue by-products. The exploitation of phenolic compounds as bioactive substances motivates scientists to explore plant raw materials more efficiently [19].

The optimization of techniques for the effective utilization of valuable natural compounds from the remaining by-products is of economic importance because the polyphenol-rich sources possess high antioxidant (AOx) potential [15]. Since involving organic solvents could provoke undesirable environmental and biological effects, water extraction of biologically active substances could be regarded as eco-friendly and consistent with the principles of ‘green chemistry’. It should be mentioned also that polyphenols are quite hydrophilic substances according to their phenolic nature [15].

The biologically active compounds of plants can be analyzed by different chromatographic techniques. High-performance thin layer chromatography (HPTLC) as an enhanced form of planar thin-layer chromatography is automated in the different steps to increase the resolution during the separation of components from their mixtures [7,20]. It is rapid, simple and easy to use. However, the optimizing mobile phase in planar chromatography is more difficult to achieve compared to the high-performance liquid chromatography (HPLC). The HPLC uses a high pressure to separate, identify, and quantify each component of an analyzed mixture, and using gradient elution significantly optimizes this process [18].

Several scientific publications demonstrate that a lot of phenolic compounds and other hydrophilic substances are unused after the process of hydrodistillation of the *Lamiaceae* representatives [16,17,18,19]. Thus, the hydrodistilled residue by-product of *Thymus vulgaris* L. herb contained up to 105.8 mg/g of rosmarinic acid (RAc) [16]. The postdistillation waste from the *Ocimum basilicum* L. leaves [17] and *Monarda fistulosa* L. herb [18] were also valuable sources of polyphenols with significant free radical scavenging potential. Sánchez-Vioque et al. [13] found that *Thymus mastichina* L. *and Lavandin* residues after the separation of essential oil were promising sources of antioxidants. Generally, there are no scientific data about obtaining or studying the residue by-product from the *O. americanum* aerial part after its hydrodistillation.

The aim of the study was to evaluate the polyphenols in the *O. americanum* dry extract (ODE) obtained from the plant’s hydrodistillation residues with the further determination of its safety and screening the pharmacological activities.

## 2. Materials and Methods

### 2.1. Plant Material

The plant raw material (herb) of the *O. americanum* was harvested from the cultivated plants in the Ternopil region (Ukraine) during the flowering stage, then shade dried and ground.

### 2.2. Chemicals

Acetic acid, formic acid, methanol, ethyl acetate, gallic acid, Folin–Ciocalteu’s reagent, AlCl_3,_ and Na_2_CO_3_ were purchased from POCH S.A. (Gliwice, Poland). The polyphenols’ standards (RAc, caffeic acid, chlorogenic acid, luteolin, apigenin, and rutin (RT)), Tween-80, carrageenan, Trolox, and 2,2-diphenyl-1-picrylhydrazyl (DPPH) were from Sigma-Aldrich. Silica gel 60 F_254_ plates for HPTLC, acetonitrile, trifluoroacetic acid and standards for HPLC (RAc, caffeic acid, ferulic acid, neochlorogenic acid, chlorogenic acid, apigenin, apigenin-7-*O*-glucoside, luteolin, L-7, acacetin-7-*O*-glucoside, catechin, RT, quercetin, hyperoside) were from Merck (Darmstadt, Germany). “Diclophenac” (produced by manufacturer “Chervona zirka”, Kharkiv, Ukraine) was purchased from the pharmacy market in Ukraine. All the solvents and reagents were of analytical grade.

### 2.3. Preparation of the O. americanum Dry Extract

The hydrodistillation method was used to separate the essential oil from the *O. americanum* herb. The ODE was obtained from its postdistillation waste. The aqueous extraction of the plant material in the process of hydrodistillation using purified water (120 min, ratio 1:15 of the plant material to water) was conducted according to the European Pharmacopoeia [14]. The obtained aqueous extract was cooled and then filtered using a paper filter. The residue of the plant raw material was then extracted again by purified water using a heater for 40 min. The obtained extract was filtrated after cooling too. Both filtrates were then combined and put in the refrigerator (for 24 h) for the sedimentation of ballast. The decanted extract was evaporated in a vacuum rotary evaporator to 1/8–1/10 of an initial volume. The concentrated extract was finally dried in a vacuum dryer.

### 2.4. Total Phenolic Content (TPC)

The TPC was studied by Folin–Ciocalteu’s method [21]. It was calculated as gallic acid equivalents (mg GAE/g of dry extract. The concentrations of reference standards (RS) of the gallic acid in the range of 0.02 to 0.14 mg/mL were used to plot the calibration curve. 0.1 mL of the extract (dissolved in purified water, 0.5 mg/mL) was mixed with 0.1 mL of Folin–Ciocalteu reagent and 1.5 mL of purified water. Then 0.3 mL of 20% solution of Na_2_CO_3_ was added. The mixtures were incubated at room temperature in darkness for 2 h. Their absorbance was measured, using spectrophotometer Hitachi U-2810 at a wavelength of 760 nm.

### 2.5. Chromatographic Analyses of the ODE

Polyphenols were identified using HPTLC analysis with a CAMAG (Switzerland) analytical system according to Shanaida et al. [7]. The extract was dissolved in methanol. The concentration of the extract in methanol was 1.0 mg/mL. A standard solution consisted of the RS dissolved in methanol (0.25 mg/mL). The mobile phase consisted of ethyl acetate, formic acid and water in a ratio of 15:1:1. The post-chromatographic derivatization was conducted by spraying a 1% solution of aluminum chloride hexahydrate as the visualization reagent for flavonoids. The analysis of spots was conducted, using UV-light (at a wavelength of = 366 nm).

The HPLC analysis of phenolic compounds was conducted with a Shimadzu HPLC-DAD system (Kyoto, Japan) using the Phenomenex Luna C18 column (250 mm × 4.6 mm × 5 µm) at a temperature of 35 °C. The gradient elution was enriched by the mixing of two mobile phases (Table 1) according to Shanaida et al. [18].

The identification of polyphenols was conducted by comparison of the retention times and the absorption spectra obtained for the reference standards and for each peak in the tested ODE as it is described in the paper [7]. The quantification of the polyphenols was achieved by measuring the peak area at the wavelengths of 280, 330 and 350 nm according to the absorption maxima of the analyzed components. The concentrations of RAc and L-7 standards for obtaining the calibration curves ranged from 10.0 to 1000.0 μg/mL. The concentrations of 1.0 to 100.0 μg/mL were applied for other polyphenols. R^2^ was at a minimum of 0.99 for all the measurements.

### 2.6. Pharmacological Activities

#### 2.6.1. Antiradical Activity against DPPH (In Vitro)

The scavenging activity of the ODE against DPPH radical was studied according to Gougoulias et Mashev [22]. 0.1 mL of the test samples (the concentrations of the ODE dissolved in methanol were in the range of 0.1–1.0 mg/mL) mixed with 1.9 mL of DPPH solution (25 μg/mL, in methanol). These mixtures were incubated for 30 min at a temperature of 22 ± 1 °C. The absorbance was evaluated at a wavelength of 760 nm, using spectrophotometer Hitachi U-2810 UV/VIS. Trolox was applied as a standard AOx. The concentration of the compounds with the AOx effect presented in the test solutions was expressed as an IC_50_ value (R^2^ = 0.9903).

#### 2.6.2. Acute Toxicity and Anti-Inflammatory Activity Studies (In Vivo)

All the experiments were carried out according to the rules of bioethics [23]. Commission on bioethics of I. Horbachevsky Ternopil National Medical University approved the research conducted *in vivo*. Approval Code: Minutes N 47 (1 June 2018).

The albino rats (200 ± 20 g) were used for the *in vivo* studies.

In the acute toxicity study, the rats of both sexes (*n* = 6 animals/group) were orally given the ODE in the doses of 500, 1500 and 5000 mg/kg body weight once a day [18,24]. The abovementioned doses of the extract were dissolved in the vehicle (1% solution of Tween-80 in purified water). The experimental animals were observed daily for up to 2 weeks. The breath, feed and water consumption, condition of wool, the character of excrements, fatality, and dynamics of body weight were analyzed daily.

The anti-inflammatory effect was observed by inducing the hind paw edema (PE) in the rats by carrageenan [18,24]. The PE was induced by the injection of 100 μL of 1% carrageenan suspension into the right paw of each animal. The rats (*n* = 6 animals/group) were orally pretreated with the extract, standard medicinal product “Diclofenac”, or vehicle 1 h before the injection of carrageenan. The rats were divided into five groups. The animals in the first group were given the 1% solution of Tween-80 in purified water (vehicle control group). The rats of the experimental groups (2–4 groups) were pretreated with the extract in the doses of 25, 50, or 100 mg/kg. The animals in the fifth group (reference drug group) were pretreated with “Diclofenac” in a dose of 8 mg/kg.

The volume of PE was measured using a plethysmograph before and 1, 3 and 6 h after the carrageenan injection. Anti-inflammatory effect (AIE) was expressed as the decreasing in PE (%) in the pretreated rats by comparison with the vehicle control group according to the formula:% AIE = [(d_c_ − d_e_)/d_c_)] × 100%
where d_e_ is the difference in paw volume in an experimental group, d_c_ is the difference in paw volume in the vehicle control group.

### 2.7. Statistical Analysis

The statistical analyzes were performed using the 13.1 version of Statistica software. The chromatographic and spectrophotometric analyzes were carried out in triplicate. The pharmacological studies were conducted in 6–10 replicates.

## 3. Results and Discussion

### 3.1. Phenolic Compounds

The spectrophotometric estimation of TPC showed a moderate amount (106.31 ± 1.97 mg GAE/g) in the studied ODE. This value was similar or higher to those found in the different *Nepetoideae* representatives [18,21,25,26]. Such environmental factors as plant growing conditions (type of soil, climate, rainfall, exposure to stressors) as well as extraction methods or solvents chosen for analysis have major effects on TPC [26,27,28]. The TPC calculated as tannic acid equivalent in the *Ocimum americanum* leaves cultivated in South-Western Nigeria was in the range of 57.76–94.0 mg GAE/g depending on the chosen solvent [9]. Majdi et al. [25] found 105 mg GAE/g of TPC in the lyophilization of the infusion from the *Ocimum basilicum cv.* ‘*Cinnamon*’ leaves. The TPC in the leaves of five Korean *Ocimum* species including *O. americanum* ranged from 66.4 to 112.3 mg GAE/g of dry extract [26]. Tufts et al. [27] found 136 mg GAE/g in the extract of Kenyan *O. americanum* leaves. Sukardi et al. [28] reported that TPC in the *O. americanum* from Indonesia ranged from 219 to 260.5 mg GAE/g of leaves extract. It should be noted that the plant raw material was macerated by researchers with 80% ethanol using a pulsed electric field as a pretreatment on the extraction [28].

The chromatographic identification of polyphenolic compounds in the ODE was firstly conducted using HPTLC. The sequence of fluorescent zones in the chromatograms of the RS and test solution scanned before and after derivatization are given in Figure 1. The most visible light blue zones (R*_f_* = 0.75) corresponding to RAc were detected on the chromatograms of the test solutions both before and after derivatization. The light blue zones corresponding to caffeic acid were revealed above the RAc spots at R*_f_* = 0.79. The weak yellow zones of RT were identified at R*_f_* = 0.10 and they were better seen after the derivatization (Figure 1b). Spraying the plates with a 1% solution of aluminum chloride causes a bathochromic shift of RT and other flavonoids, i.e., a shift of the absorption maximum of the molecule in the area of longer wavelengths. Thus, the derivatization enhances the yellow fluorescence of flavonoids. Several unidentified spots in different fluorescent shades of yellow and blue colors which could be considered as flavonoids or hydroxycinnamic acids, respectively, were identified in the ODE chromatograms in different places. Similar ‘chromatographic fingerprints’ were obtained by researchers after the extraction of *O. americanum* herb with methanol [7]. Generally, the obtained polyphenols can represent a pattern of biologically active compounds revealed in the ODE.

The detailed component analysis of phenolic compounds in the ODE was further conducted using the HPLC method (Table 2, Figure 2). The quantification of hydroxycinnamic acids (RAc, caffeic acid, neochlorogenic, chlorogenic, ferulic) and flavonoids (catechin, RT, hyperoside, quercetin, apigenin-7-*O*-glucoside, acacetin-7-*O*-glucoside, L-7, apigenin, luteolin) was performed by the HPLC analysis.

As could be seen from the abovementioned results, a lot of polyphenols were revealed in the ODE by the HPTLC and HPLC methods. They were recovered from the postdistillation residues that otherwise would be lost. Among them were hydroxycinnamic acids, flavones and flavanols possessing good anti-inflammatory, AOx and immunomodulatory activities [29,30,31,32].

RAc is the major phenolic compound of the studied ODE as well as many species of the *Nepetoideae* subfamily of the *Lamiaceae* family which are phylogenetically close to the *tribe* *Ocimeae* Dumort. [18,25,29,33,34]. Thus, Majdi et al. [25] detected 41.0 mg/g of RAc in the lyophilized extract obtained after drying the *Ocimum basilicum cv.* ‘*Cinnamon*’ infusion. 10.97 mg/g of RAc were found in the leaf extract of *O. americanum,* which were collected in India [29]. Farag et al. [33] assessed the differences in presence of polyphenols in the methanol extracts of leaves gathered from four Egyptian species (*Ocimum* (*O. basilicum*, *O. africanum*, *O. americanum* and *O. minimum*) using HPLC/DAD. RAc was identified as a key phenolic compound of these *Ocimum* leaves and it was followed by other cinnamic acid derivatives (monomers, dimers and trimers) [33]. The findings of Toma et al. [31] revealed that such hydroxycinnamic acid as caffeic one which follows the RAc in the studied ODE also possess noticeable AOx properties. Toma et al. [31] found that caffeic acid can inhibit reactive oxygen species which are observed in inflammation processes.

The therapeutic properties of flavonoids detected in the ODE are also very significant [34,35]. 10.9 mg/g of RT was revealed by Pandey et al. [29] in the Indian *O. americanum* leaves’ extract that correlates with the results obtained by us in the ODE (11.20 mg/g). The pharmacokinetic studies of Habtemariam and Belai [34] showed the ability of RT to deliver more amounts of bioactive aglycon quercetin in the digestive tract. Quercetin can significantly suppress releasing pro-inflammatory mediators [34]. Park and Song [35] revealed the noticeable anti-inflammatory properties of flavone luteolin and its derivatives.

### 3.2. Evaluation of Bioactive Properties

Oxidative stress is regarded as the manifestation of an imbalance between releasing reactive oxygen or nitrogen species in the body and their elimination [36]. If the AOx system cannot neutralize them, free radicals provoke the development of chronic inflammation which cause such diseases as insulin resistance, cardiovascular ailments, neurodegeneration, arthritis, cancer, obesity, among other pathological conditions [36,37].

The beneficial pharmacological activities of polyphenols as natural antioxidants give rise to prophylaxis or cure of the abovementioned human diseases. There is no doubt that synthetic antioxidants possess plenty of side effects such as intoxication, liver damage, or carcinogenesis. Natural phenolic compounds containing two or more OH-groups can donate their protons to reactive oxygen species [29,30]. Luo et al. [30] found that RAc as the major component of ODE, which contains four OH-groups, demonstrates significant AOx potential.

The anti-DPPH assay was used to study the AOx effect of the investigated ODE. The developed ODE had moderate AOx property with an IC_50_ value of 0.298 mg/mL. Song et al. [26] found that the *O. americanum* extract (1.0 mg/mL) was capable to scavenge the DPPH free radical by 55.2%. The methanol extract obtained by Oyedemi et al. [38] from the Nigerian *O. americanum* leaves demonstrated strong anti-DPPH activity (IC_50_ = 0.147 mg/mL). The antioxidant activity of the *O. americanum* extracts, which was evaluated by the oxygen radical absorbance capacity, strongly correlated with the TPC determined by Folin-Ciocalteu’s method [27]. It should be mentioned that comparing the results of the AOx studies is complicated as the researchers use the different solvents and variety of experimental models besides DPPH assays [12,25].

The acute toxicity study of the ODE was recorded for two-weeks post the treatment of the experimental animals. It was indicated that the ODE in the tested doses of 500, 1500 and 5000 mg/kg did not provoke death or any side effects in rats. Thus, up to 5000 mg/kg, the ODE could be considered safe.

The PE test is most commonly used in the evaluation of the efficacy of anti-inflammatory medicines that act by inhibiting the acute inflammation mediators [18,24]. After carrageenan injection, the volume of the animal paw increased as edema developed. That indicated the inflammatory effect of the carrageenan. The found anti-inflammatory effect of the ODE was similar to “Diclofenac” as a standard medicinal product. The tested ODE significantly diminished the PE in the rats caused by the carrageenan at 3 h of the induced inflammation (Table 3).

Thus, the tested ODE showed a noticeable anti-inflammatory effect compared to the vehicle control group in the PE test. The anti-inflammatory effect of the ODE was manifested dose-dependently. It was observed that the dose of 25 mg/kg did not show the robust anti-exudative effect whilst the highest dose (100 mg/kg) decreased the PE by 31.71% (at the 3 h). However, it was less intensive compared to the reference drug “Diclofenac”. At the 6 h of the observation, the effect of the ODE at a dose of 100 mg/kg did not demonstrate the significant difference from the “Diclofenac”.

Our results concerning the anti-inflammatory properties of the ODE were similar to the obtained results by researchers of the *Nepetoideae* subfamily. Thus, Yousuf et al. [39] found that the *Mentha spicata* (*Lamiaceae*) methanol extract exhibited promising anti-inflammatory properties in the carrageen induced PE experiment. The dose of 250 mg/kg body weight demonstrated decreasing paw volume by 42.58% at 6 h of the experiment [39]. Recently, Genfi et al. [10] supposed that the hepatoprotective influence of *Ocimum americanum* aqueous extract was due to the ability to inhibit the proinflammatory cytokines.

It can be supposed that the found anti-inflammatory property of the ODE, as well as its AOx effect, was due to the high amount of RAc, L-7, RT and other bioactive polyphenols. It should be mentioned that inflammation is closely related to oxidation as the damage of cells by free radicals leads to the development of inflammation. Thus, the anti-inflammatory property of the developed ODE in the range of doses of 25–100 mg/kg could be due to high contents of RAc and other polyphenols inhibiting the inflammation processes [40,41,42]. It should be noted that the aqueous extracts of *Salvia officinalis* leaves demonstrated anti-inflammatory and analgesic effects in much higher doses (up to 1000 mg/kg) [43]. Hussain et al. [36] supposed that different polyphenols can modulate the activity of cyclooxygenase and lipoxygenase, which leads to reduced synthesis of arachidonic acid, prostaglandins, and leukotrienes as the key mediators of inflammation. Polyphenols can repress macrophages as the noticeable players in the inflammatory pathway [37]. Luo et al. [30] found the promising anti-inflammatory properties of hydroxycinnamic RAc Rocha et al. [44] revealed that administration of pure RAc (25 mg/kg) diminished the PE in animals by over 60%. Park and Song [35] found that flavonoid L-7, the second major component of the ODE, can inhibit the inflammatory response induced by lipopolysaccharide. Habtemariam and Belai [34] demonstrated that RT and especially its aglycon quercetin also effectively suppressed the inflammatory cytokines.

## 4. Conclusions

The developed ODE is a residue by-product from the *O. americanum* herb after hydrodistillation recovery that otherwise would be lost. It possesses the AOx as well as anti-inflammatory effects. The performed chromatographic studies revealed the presence of many polyphenols in the ODE. The high amounts of RAc, L-7 and RT were found using HPLC. The moderate TPC measured spectrophotometrically correlated with the temperate AOx effect against DPPH. *In vivo* studies showed that administration of the ODE at the doses of 500–5000 mg/kg to the rats did not reveal any toxic reactions that indicate its safety. Dose-dependent anti-inflammatory activity of the ODE was found. The obtained results indicate that the developed dry extract is a promising natural source of polyphenolic compounds. The ODE could be regarded as a prospective source of herbal drugs and food additives for the treatment or prevention the diseases caused by oxidative stress and inflammation.

## Figures and Tables

**Figure 1 foods-10-03063-f001:**
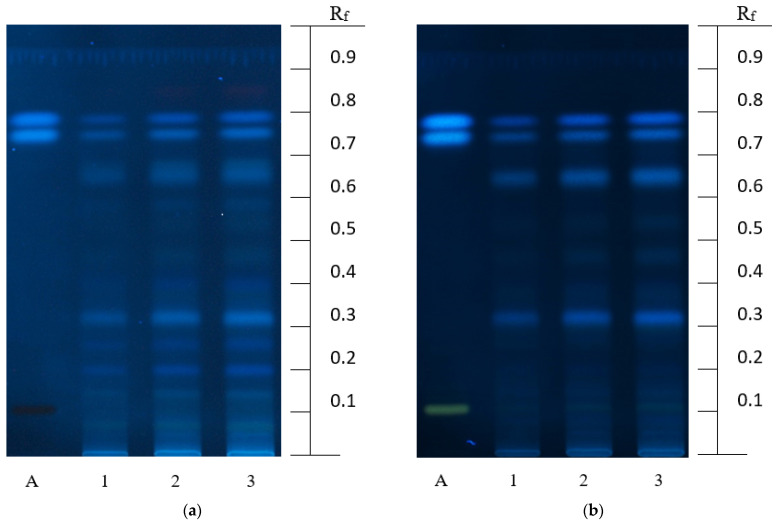
Typical HPTLC chromatograms of the ODE test solution (1–3) and RS of polyphenol (A: RT, RAc, and caffeic acid with increasing R*_f_*) before (**a**) and after (**b**) derivatization at λ = 366 nm.

**Figure 2 foods-10-03063-f002:**
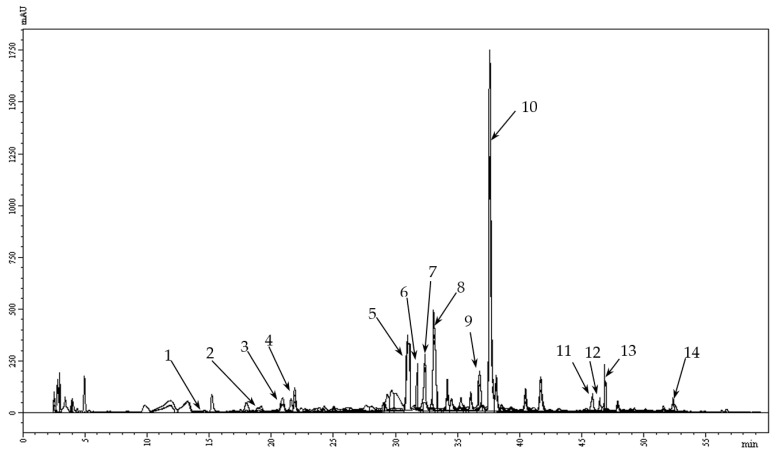
The typical HPLC chromatogram of polyphenols in the ODE (at λ = 330 nm): 1—neochlorogenic acid; 2—catechin; 3—chlorogenic acid; 4—caffeic acid; 5—RT; 6—hyperoside; 7—ferulic acid; 8—L-7; 9—apigenin-7-*O*-glucoside; 10—RAc; 11—acacetin-7-*O*-glucoside; 12—quercetin; 13—luteolin; 14—apigenin.

**Table 1 foods-10-03063-t001:** Gradient of the mobile phases in HPLC-analysis [18].

Time (min) after Injection of a Sample	Mobile Phase A (Vol, %)	Mobile Phase B (Vol, %)
0–5	95	5
5–35	95 → 75	5 → 25
35–40	75	25
40–60	75 → 50	25 → 50
60–65	50 → 20	50 → 80
65–70	20	80
70–85	95	5

**Table 2 foods-10-03063-t002:** Contents of polyphenols in the ODE evaluated by the HPLC.

Compound	Retention Time, min	Content, mg/g of Dry Extract
Neochlorogenic acid	14.8	0.31 ± 0.01
Catechin	19.5	0.90 ± 0.04
Chlorogenic acid	20.4	0.95 ± 0.03
Caffeic acid	21.6	4.13 ± 0.11
Rutin	30.9	11.20 ± 0.26
Hyperoside	31.6	6.34 ± 0.12
Ferulic acid	32.3	8.21 ± 0.09
Luteolin-7-*O*-glucoside	33.1	17.22 ± 0.49
Apigenin-7-*O*-glucoside	36.8	5.64 ± 0.12
Rosmarinic acid	37.8	78.70 ± 1.13
Acacetin-7-*O*-glucoside	45.8	3.51 ± 0.08
Quercetin	46.6	2.21 ± 0.06
Luteolin	47.0	7.82 ± 0.14
Apigenin	52.4	1.94 ± 0.05

**Table 3 foods-10-03063-t003:** Anti-exudative effect of the ODE on paw edema in rats induced by carrageenan.

Treatment	Dose (mg/kg)	Increase in Paw Oedema					
		After 1 h		After 3 h		After 6 h	
		Diff	% AIE	Diff	% AIE	Diff	% AIE
Control	-	0.34 ± 0.03	-	0.41 ± 0.02	-	0.38 ± 0.02	-
ODE	25	0.29 ± 0.02 ^2^	14.71	0.33 ± 0.03 ^2^	19.5	0.34 ± 0.02 ^1,2^	10.53
	50	0.27 ± 0.03 ^2^	23.53	0.30 ± 0.02 ^1,2^	26.83	0.31 ± 0.01 ^1^	18.42
	100	0.24 ± 0.01 ^1,2^	29.41	0.28 ± 0.01 ^1,2^	31.71	0.29 ± 0.02 ^1^	23.68
Diclofenac	8	0.18 ± 0.02 ^1^	47.05	0.21 ± 0.03 ^1^	48.78	0.29 ± 0.01 ^1^	23.68

Diff—difference in paw volume of the rats before and after injection of carrageenan; ^1^—significantly different compared to untreated control (*p* ≤ 0.05); ^2^—significantly different compared to Diclofenac group (*p* ≤ 0.05).

## Data Availability

Data are contained within the article.

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
