# Peer review of "Phytochemical and Pharmacological Evaluation of the Residue By-Product Developed from the Ocimum americanum (Lamiaceae) Postdistillation Waste"

_foods, 2021, doi:10.3390/foods10123063_

Round 1
Reviewer 1 Report
The manuscript "Phytochemical and Pharmacological Evaluation of the Residue By-Product Developed from the Ocimum americanum (Lamiaceae) Postdistillation Waste" reports an exhaustive characterization of the main phytochemicals in a dry extract obtained from the postdistillation waste of the American basil herb.
This is a fairly well-written manuscript, requiring moderate editing of English language. The chromatographic, spectrophotometric and pharmacological studies have been properly conducted.
My concerns go to the lack of potential interest for the hydrodistillation industry, or even for readers, as this is mainly a descriptive manuscript.
Moreover, concluding that "the O. americanum hydrodistilled residue by-product is as a promising source of biologically active compounds with the significant antioxidant and anti-inflammatory effects" does not present any innovation or advantage for the hydrodistillation industry.
Author Response
Dear Reviewer!
Thank you a lot for your remarks! We conducted the thorough revision of the manuscript according to them.

Reviewer 2 Report
The authors need to follow the following instructions to improve this manuscript.
- Line 9: Please check the line alignment.
- Line 16,40: Basil word writing should check. One place small b another place capital B.
- Abstract: The authors avoid using abbreviations in the Abstract, especially ODE, RAc, RT, HPTLC, HPLC, TPC, PE, AOx.
- Line 17: Abbreviation ODE. What is the meaning of ODE?
- Line 34: The authors short avoid using etc. and follow the entire manuscript.
- Line 36: Reference numbers. Should minimize the gap between 2 numbers and follow the entire manuscript.
- Line 132: Check the reference writing of 22.
- Line 169: (Rf=0.75)- Check the space and follow the entire manuscript.
- Table 1: Write the full name of RT, L-7, RAc.
- Table 2: Check the font.
- References should check clearly. Check the Journal rules and regulations. Before submission, it is mandatory to check the journal reference writing style.
- The authors used 52 references—too many. I think the authors should use recent and relevant ones, skip the old ones.
- The manuscript should check the English by professionals or editing companies.
Author Response

(The authors gave the same response as above.)

Reviewer 3 Report
The valuation of agricultural and food by-products is of great interest, and this study is therefore part of this important trend.
First, it appears that this ms has been submitted in a revised form, but I would like to point out that I was not involved in a previous peer review round for this ms. So don't be surprised if some of my comments are redundant with previous comments that could have been made.
The presentation of the phytochemical characterization does not follow a logical progression in my opinion. It would be preferable to start with a global assay such as the determination of total phenolic and then to continue with more precise analyzes such as those proposed by the authors in HPTLC and HPLC.
The interest which the authors draw from these last two analyzes (i.e., HTPLC and HPLC), redundant in my opinion (except justification of the authors), is not very clear. It would be good for the reader to clearly explain the interest of these two methods, their respective limits, their complementarities and what they bring each of them for the present study...
It is important to indicate in the legend of FIG. 2 the wavelength at which the compounds presented were detected (as the authors did for FIG. 1).
The authors themselves indicate that many studies of the same type have been carried out both on different species of basil and also on Lamiaceae. It is important to state much more clearly why the present results are new.
In view of the quantity of experimental approach and their technical difficulties, it seems to me that a short communication would be more appropriate than a full paper. Here again, it seems important to me to justify the need to publish these results (all in all rather preliminary, although I do not doubt their importance that the authors will be able to highlight during the revision process).
Author Response

(The authors gave the same response as above.)

Reviewer 4 Report
The manuscript reports a chemical and pharmacological evaluation of residues of the Ocimum americanum hydrodistillation. It shows a good level of data but they were not presented and discussed adequately. Authors should revise text. Some parts of the discussion appear not linked to the obtained results; they should make an effort to connect references reported to the results. Some comments are reported below about some points and parts of the manuscript.
Introduction
The introduction describes the scientific problem but, if authors agree, I suggest change the order of some parts. The discussion at lines 64-68 could be move after discussion of distillation of line 57. In this way, there is a continuity of issues. In the same way, phrases after that (line 57) should be slightly adjust.
Material and methods
2.3. Preparation of the O. americanum dry extract
The hydrodistillation uses water. The residual water of distillation process could contain phenol substances. Have authors verified if the water they used to distillate contained these substances? Have they recovered the residual water? Could they explain?
2.4. Chromatographic analyses of the ODE
Authors could briefly describe the gradient elution even if they reported the reference about the method.
Authors should briefly describe the post-chromatographic derivatisation. For example, the AlCl3 solution was spry on the plate?
2.6.1. Antiradical activity against DPPH
Have authors thermostatted the sample during the 30 minutes reaction? The isothermal condition is very important to obtain a significant DPPH antioxidant test.
Discussion
Lines 169-171. It is not clearly identified the different spots from the Figure 1. The terms “most visible light blue”, “light blue” and “weak yellow” zones did not well define the different spots on Figure (probably it could be a print problem). I suggest to insert into Figure some signs, or define that text refers to figure 1b, the only of two where the yellow spot is clearly visible.
Figure 1. It is not clear from the discussion why authors have reported the figure 1. Did they want underline the differences between compounds before and after derivatization? If so, they should discuss this aspect into text. Otherwise, the figure is not useful.
Table 1 and text (Material and Methods, Discussion). Authors should define in table (but also in text when reported) if the concentration of the single compound is reported as mg of gram of distillation residue, extract and if it is reported as residue, it should be specify as dry matter.
Lines 186-202. The discussion reported is not clearly linked with the results of the polyphenol content (table 1 and figure 2). Authors reported a short survey about other researches, not all of them are well linked with phenols of O. americanum. I suggest modifying this part making it as much as possible connected with text. The references reported are hardly to compare with data reported in table 1 (different spices, extraction method, and vegetable part). Same consideration could be done for discussion at lines 214-228. In conclusion, of this comment, I underline that the list of researches and phenol (TPC or single component) contents is not useful. Authors should try to give a strong relation with the aim of their work: recovery of fraction with biological activity from residues that otherwise they would be lost.
Lines 240 and subsequent. Authors should better specify the unit of measure for IC50. The amount of mg is related to extract? I suggest to specify into discussion or (better) into material and methods part.
Author Response

(The authors gave the same response as above.)

Round 2
Reviewer 3 Report
The revision has been conducted by the Authors.
Author Response
Dear Reviewer!
Thank you very much for the approval of the adjustments made by us
Reviewer 4 Report
The manuscript is improved respect previous version but authors did not explain some aspects.
- In my previous revision I have asked, respect paragraph 2.3., if the water used to distillate contained phenol compounds. And if authors have recovered the residual water. They didn’t explain.
- (2nd version lines 188-195 and 268-274) As I underlined in my previous revision, authors have reported a short survey about other researches, not all of them are well linked with phenols of O. americanum. I suggest to maintain only those related to the O. americanum.
Author Response
Dear Reviewer!
Thank you very much for the valuable remarks! We hope that the text of manuscript is better now.

This manuscript is a resubmission of an earlier submission. The following is a list of the peer review reports and author responses from that submission.
Round 1
Reviewer 1 Report
In its present form, it is impossible for me to understand the meaning of the work. To judge the scientific value of this paper, the manuscript needs a thorough review of the English language.
Reviewer 2 Report
The manuscript by Jasicka-Misiak et al. is a fairly well written manuscript. This is essentially a descriptive manuscript, aiming at evaluating polyphenols of O. americanum herb, along with screening of its pharmacological activities.
My main criticism is the lack of originality, as many papers have been published reporting promising sources of biologically active compounds with the significant antioxidant and anti-inflammatory effects. In addition, I have some concerns with the chromatographic analyses, which are not well described (lack of information on the standards and quantification method). Mass spectrometric analysis should be conducted, in order to identify the hydroxicinnamic acids and flavonoids.
In conclusion, the current manuscript does not fit into the high standards of the FOODS journal.
L: 297: "The administration of the ODE at the doses of 500–5000 mg/kg to rats did not revealed any toxic reactions that indicates its safety."
L 245: "It was indicated that the ODE in teste doses 0.5 g/kg, 500 g/kg and 5000 g/kg did not provoke death or any side effects in the rats. Thus, up to 5 g/kg, the ODE could be considered as a safe."